Neglected Tropical Diseases

# Expression of interleukin-4 in schistosomiasis is influenced by age

Tresford Lwanga[1‡], Lweendo Muchaili[1,2‡], Gift C. Chama[2], Lukundo Siame[1,2,3], Cornelius Simutanda[3], Situmbeko Liweleya[2], Bislom C. Mweene[1,2], Sydney Mulamfu[1,2], Sepiso K. Masenga [1,2,3]*

1 Department of Pathology, Division of Integrated Sciences, Livingstone Center for Prevention and Translational Science, Livingstone, Zambia, 2 Department of Pathology and Microbiology, Mulungushi University, School of Medicine and Health Sciences, Livingstone, Zambia, 3 Department of Medicine, Division of Integrated Sciences, Livingstone Center for Prevention and Translational Science, Livingstone, Zambia

‡ First co-authorship.
* sepisomasenga@gmail.com, sepisomasenga@lcpts.org

## Abstract

### Background

Schistosomiasis, a Neglected Tropical Disease (NTD), remains a significant public health issue, particularly in sub-Saharan Africa. Despite various interventions, the disease persists, with a considerable burden on affected populations. This study aimed to characterize the hematological and immunological profiles associated with Schistosoma infection in pediatric and adult populations in Mulobezi district of Zambia.

### Methods

This was a cross-sectional study, which included participants aged 5–55 years, carried out in Mulobezi District in the Western Province of Zambia. The sample size was 143, participants were stratified into children (<15 years) and adults (≥15 years). Schistosomiasis diagnosis was confirmed through urine microscopy using filtration methods, while full blood count and cytokine (IL-4, IL-10) analysis were performed on blood samples. A Multivariate logistic regression was used to identify factors independently associated with infection. Statistical analyses were conducted using two-sided tests to assess the significance of the observed differences between groups.

### Results

Out of the 143 participants, 56 (39.2%) had schistosomiasis. In adjusted models, IL-4 showed age-dependent associations: significant in adults (AOR: 1.007, 95% CI: 1.003-1.012, p = 0.001) but not children (AOR: 1.001, 95% CI: 0.999-1.003, p = 0.072). Lymphocyte counts were elevated in both children (AOR: 1.94, 95% CI: 1.07-3.54, p = 0.029) and adults (AOR: 2.96, 95% CI: 1.19-7.36, p = 0.020).

**Data availability statement:** All relevant data are within the paper and its Supporting Information files.

**Funding:** The author(s) received no specific funding for this work.

**Competing interests:** The authors have declared that no competing interests exist.

Eosinophils (children: AOR: 13.94, 95% CI: 0.24-811.21, p = 0.204; adults: AOR: 12.27, 95% CI: 0.73-207.13, p = 0.082) and IL-10 lost significance after adjustment.

## Conclusion

This study highlights IL-4 and lymphocyte counts as potential immunomarkers for schistosomiasis infection. The age-specific IL-4 association suggests differential immune activation patterns, while persistent lymphocyte elevation across both groups indicates sustained adaptive immune engagement.

## Author summary

Schistosomiasis, a parasitic disease affecting millions in sub-Saharan Africa, triggers complex immune responses that vary with age. This study, conducted in Zambia's Mulobezi District, analyzed 143 participants (aged 5–55) to identify age-specific immune markers linked to infection. Using urine microscopy for diagnosis and blood tests to measure cytokines and blood cells, we found that 39% of participants had schistosomiasis.

Key findings revealed that interleukin-4 (IL-4), a protein driving immune defense against parasites, was strongly associated with infection in adults but not children. Lymphocytes, white blood cells critical for adaptive immunity, were elevated in both age groups, suggesting sustained immune activation. Traditional markers like eosinophils (allergy-related cells) and anti-inflammatory IL-10 lost significance after adjusting for other factors, highlighting their limited diagnostic specificity.

The age-dependent role of IL-4 implies that chronic exposure reshapes immune responses, potentially worsening tissue damage in adults. Persistent lymphocyte elevation across ages underscores their importance in combating infection. These findings advance our understanding of how immune mechanisms evolve with age and exposure, offering insights for improved diagnostics and tailored treatments. By identifying IL-4 and lymphocytes as robust biomarkers, this study paves the way for better surveillance strategies in high-risk regions, emphasizing the need for age-specific approaches in managing schistosomiasis.

## Introduction

Schistosomiasis is a major public health challenge in sub-Saharan Africa which has over 90% of the global burden concentrated in the region [1]. Previous estimates put the burden of schistosomiasis in Zambia at 2 million people with the majority of them in rural communities with limited access to clean water and sanitation [2]. Schistosomiasis manifests through a number of host-parasite interactions, thereby triggering

distinct immunological responses that vary by age and exposure duration. While the typical T helper 2 ($T_H2$)-skewed immune response to schistosomiasis is well-documented, the age-specific dynamics of key immunological markers, principally involving IL-4 and lymphocyte profiles, remain poorly characterized in endemic Zambian populations [3,4].

The immunopathology of schistosomiasis is mainly driven by host immune responses to parasite eggs, with cytokines playing important roles in modulating disease outcomes [5]. IL-4 and IL-10 have well-established and complementary roles in schistosomiasis immunobiology [6–8]. IL-4 is central in orchestrating the $T_H2$-mediated immune response, promoting eosinophil activation and IgE production, both of which are hallmarks of schistosomiasis-associated granuloma formation [9,10]. On the other hand, IL-10 serves as an anti-inflammatory cytokine that counterbalances excessive immune activation, limiting tissue damage caused by the immune response [11]. These cytokines thus represent key immunological mediators in the host-parasite interaction and provide valuable insights into disease progression and control strategies.

Current diagnostic methods rely primarily on the parasitological detection of eggs in urine and stool, which has limited sensitivity in low-intensity infections [12]. Immunological biomarkers could complement existing tools by providing deeper insight into infection status and immune activation patterns [5]. However, most studies have focused on eosinophilia as a hallmark of infection despite its nonspecific nature [13–16]. Few investigations have systematically evaluated IL-4, mainly secreted by $T_H2$ lymphocytes and other lymphocyte subsets across pediatric and adult populations in high-transmission settings.

The present study addresses these gaps by analyzing hematological and cytokine profiles in schistosomiasis-infected individuals from rural Zambia stratified by age. We focus specifically on IL-4 and lymphocyte counts as potential immunomarkers while also examining why traditional markers like eosinophils and IL-10 may lack diagnostic specificity. Our findings aim to refine the understanding of age-dependent immune responses to schistosomiasis and identify robust biomarkers that could improve surveillance in endemic areas.

## Results

### Basic characteristics of study participants

The study enrolled a total of 143 participants, and the sample was stratified into two age groups to investigate age-related differences in immunological and hematological factors associated with schistosomiasis. Of the total participants, 65 (45.5%) were classified as pediatric (<15 years) and 78 (54.5%) as adults (≥15 years), Table 1 and 2. A total of 56 (39.2%) participants had schistosomiasis. Among pediatric participants, the median eosinophil count was $0.14 \times 10^3$/μL, lymphocyte count was $2.4 \times 10^3$/μL, and hemoglobin (Hb) concentration was 11.5 g/dL. The median levels of the regulatory cytokines were 56.1 pg/ml for IL-4 and 29.9 pg/ml for IL-10. Adult participants showed a lower median eosinophil count of $0.10 \times 10^3$/μL and lymphocyte count of $2.3 \times 10^3$/μL but a higher median hemoglobin level of 12.7 g/dL. The median concentration of IL-4 was 57.7 pg/ml, while IL-10 was 25.8 pg/ml in this group.

### Relationship between schistosomiasis and hematological and immunological parameters

**Immunological markers.** In pediatric participants (<15 years), those with schistosomiasis had significantly higher IL-4 levels (median: 279.7 pg/mL, IQR: 188.7–444.7) compared to uninfected counterparts (median: 28.5 pg/mL, IQR: 14.9–100), p < 0.0001, Table 2. IL-10 levels were also significantly higher in the infected participants (median: 26 pg/mL, IQR: 18.05–43.29) when compared to their uninfected counterparts (median: 26 pg/mL, IQR: 19.96–50.93), p < 0.0001.

Similarly, participants in the adult population (≥15 years) also had significantly elevated IL-4 levels in infected individuals (median: 392.8 pg/mL, IQR: 259.3–853.3) compared to their uninfected counterparts (median: 41.35 pg/mL, IQR: 21.88–56.1), p < 0.0001. IL-10 levels followed a similar trend, with significantly higher values in schistosomiasis-infected individuals (median: 27.41 pg/mL, IQR: 19.36–64.29) than in uninfected individuals (median: 32.51 pg/mL, IQR: 20.34–51.51), p < 0.0001.

**Table 1.** Hematologic and immune profiles in relation to Schistosomiasis infection in pediatric participants.

| Variable | Schistosomiasis Present (<15 years) | Schistosomiasis Absent (<15 years) | p-value |
|---|---|---|---|
| n (%) | 31 (47.7%) | 34 (52.3%) | |
| Sex | | | |
| **Male** | 12 (38.7%) | 19 (62.3%) | 0.1662 |
| **Female** | 19 (55.9%) | 15 (44.1%) | |
| IL-4 pg/mL, IQR | 279.7 (188.7, 444.7) | 28.5(14.9-100) | **<0.0001** |
| IL-10 pg/mL, IQR | 26. (18.05-43.29) | 26 (19.96-50.93) | **<0.0001** |
| Eosinophils (×10³/µL) | 0.14 (0.08-0.7) | 0.09 (0.05-0.12) | **0.0047** |
| HB (g/dL) | 11.4 (10-11.8) | 12.6 (14.1-2.9) | **0.0089** |
| Neutrophils (×10³/µL) | 1.59 (2.42-1.36) | 1.1 (2.1-1.29) | 0.0784 |
| Lymphocytes (×10³/µL) | 3.14 (2.35-4.6) | 1.73 (1.51-2.4) | **<0.0001** |
| Monocytes (×10³/µL) | 0.58 (0.16-0.9) | 0.23 (0.12-0.5) | **<0.0001** |

Abbreviations: IL 4, Interleukin 4; IL-10, Interleukin 10; HB, Hemoglobin; IQR, Interquartile range.

**Table 2.** Hematologic and immune profiles in relation to schistosomiasis infection in adult participants.

| Variable | Schistosomiasis Present (≥15 years) | Schistosomiasis Absent (≥15 years) | p-value |
|---|---|---|---|
| n (%) | 25 (32%) | 53 (68%) | |
| Sex | | | 0.8197 |
| **Male** | 6 (30%) | 14 (70%) | |
| **Female** | 19 (32.8%) | 39 (67.2%) | |
| IL-4 pg/mL, IQR | 392.8 (259.3–853.3) | 41.35 (21.88–56.1) | **<0.0001** |
| IL-10 pg/mL, IQR | 27.41 (19.36–64.29) | 32.51 (20.34–51.51) | **<0.0001** |
| Eosinophils (×10³/µL) | 0.44 (0.09–0.98) | 0.11 (0.06–0.28) | **0.0046** |
| HB (g/dL) | 11.2 (9.6–12.9) | 12.2 (11.5–13.3) | **0.0115** |
| Neutrophils (×10³/µL) | 1.29 (1.04–1.76) | 1.12 (1.01–1.71) | 0.822 |
| Lymphocytes (×10³/µL) | 2.52 (2.4–4.5) | 2.14 (1.78–2.54) | **0.0047** |
| Monocytes (×10³/µL) | 0.4 (0.2–0.96) | 0.2 (0.14–0.34) | 0.0537 |

Abbreviations: IL 4, Interleukin 4; IL-10, Interleukin 10; HB, Hemoglobin; IQR, Interquartile range.

**Hematological parameters.** Among pediatric participants, schistosomiasis infection was associated with higher eosinophil counts (median: $0.14 \times 10^3$/µL, IQR: 0.08–0.7) compared to uninfected individuals (median: $0.09 \times 10^3$/µL, IQR: 0.05–0.12), p = 0.0047, Table 2. Similarly, lymphocyte (median: $3.14 \times 10^3$/µL, IQR: 2.35–4.6 vs. $1.73 \times 10^3$/µL, IQR: 1.51–2.4; p < 0.0001) and monocyte counts (median: $0.58 \times 10^3$/µL, IQR: 0.16–0.9 vs. $0.23 \times 10^3$/µL, IQR: 0.12–0.5; p < 0.0001) were significantly elevated in infected individuals. Neutrophil counts did not differ significantly between groups (p = 0.0784). Additionally, pediatric participants with schistosomiasis had significantly lower hemoglobin levels (median: 11.4 g/dL, IQR: 10–11.8) compared to their uninfected counterparts (median: 12.6 g/dL, IQR: 14.1–2.9), p = 0.0089.

In the adult population, eosinophil counts were also significantly higher in the infected (median: $0.44 \times 10^3$/µL, IQR: 0.09–0.98) when compared to the uninfected (median: $0.11 \times 10^3$/µL, IQR: 0.06–0.28). p = 0.0046. Similarly, lymphocyte counts were also significantly higher in the schistosomiasis-infected participants (median: $2.52 \times 10^3$/µL, IQR:

2.4–4.5) when compared to the uninfected (median: 2.14 × 10³/μL, IQR: 1.78–2.54). p = 0.0047. Adult participants also had significantly reduced hemoglobin levels (median: 11.2 g/dL, IQR: 9.6–12.9) relative to their uninfected counterparts (median: 12.2 g/dL, IQR: 11.5–13.3) (p = 0.0115). On the other hand, monocyte counts showed a non-significant increase, p = 0.0537, while neutrophil levels remained comparable between groups, p = 0.822.

## Univariate and multivariate analysis of factors associated with schistosomiasis

**Pediatric participants (<15 years).** In the univariate analysis, lymphocyte count (OR: 2.505, 95% CI: 1.45–4.31, p = 0.0009) and monocyte count (OR: 6.006, 95% CI: 1.40–25.71, p = 0.0157) showed significant positive associations with schistosomiasis infection, Table 3. Eosinophil count also demonstrated a marginal association (OR: 17.7, 95% CI: 1.12–279.17, p = 0.0407). However, IL-4 levels (p = 0.0611), IL-10 levels (p = 0.24), hemoglobin (p = 0.0521), and gender (p = 0.1684) were not statistically significant predictors in the crude analysis.

After adjusting for potential confounders in the multivariate model, only lymphocyte count remained significantly associated with schistosomiasis (AOR: 1.944, 95% CI: 1.068–3.536, p = 0.0294). The associations for eosinophils (AOR: 13.939, p = 0.2038), monocytes (AOR: 2.166, p = 0.4453), and hemoglobin (AOR: 0.830, p = 0.1684) were no longer statistically significant.

**Adult participants (≥15 years).** In the univariate analysis, IL-4 levels (OR: 1.008, 95% CI: 1.00–1.01, p < 0.0001), eosinophil count (OR: 11.10, 95% CI: 2.60–47.43, p = 0.0011), lymphocyte count (OR: 1.89, 95% CI: 1.15–3.12, p = 0.0118), and monocyte count (OR: 3.52, 95% CI: 1.14–10.83, p = 0.0283) were significantly associated with schistosomiasis, Table 4. Lower hemoglobin levels also showed an association (OR: 0.76, 95% CI: 0.59–0.96, p = 0.0238), while gender (p = 0.8197) and IL-10 levels (p = 0.3231) were not significant predictors.

In the adjusted multivariate model, IL-4 (AOR: 1.007, 95% CI: 1.003–1.012, p = 0.0011) and lymphocyte count (AOR: 2.956, 95% CI: 1.186–7.364, p = 0.0199) remained independently associated with schistosomiasis. However, eosinophil count (AOR: 12.265, p = 0.0822), monocyte count (AOR: 0.740, p = 0.7687), and hemoglobin (AOR: 0.734, p = 0.1043) lost statistical significance after adjustment.

## Discussion

The findings of this study provide valuable insights into the complex immunological and hematological responses to Schistosoma infection across different age groups in rural Zambia. Our multivariate analysis revealed several key patterns that need careful consideration in the context of existing literature on schistosomiasis immunopathogenesis. The most interesting findings are on the differential expression of IL-4 between pediatric and adult populations, along with the consistent

Table 3. Immunological and hematological factors associated with schistosomiasis in pediatric participants.

| Variable | Crude analysis | | p-value | Adjusted analysis | | p-value |
|---|---|---|---|---|---|---|
| | OR | 95%CI | | AOR | 95%CI | |
| Sex | 2.005 | 0.745-5.398 | 0.1684 | **1.289** | **0.317-5.257** | 0.7224 |
| IL 4 (pg/ml) | 1.001 | 0.999-1.003 | < 0.0611 | 1.001 | 0.999-1.003 | 0.0715 |
| IL 10 (pg/ml) | 0.986 | 0.963-1.009 | **0.24** | | | |
| Eosinophils count (μL) | 17.7 | 1.12-279.17 | **0.0407** | 13.939 | 0.239-811.213 | 0.2038 |
| HB (g/dl) | 0.795 | 0.63-1.00 | 0.0521 | 0.830 | 0.638-1.081 | 0.1684 |
| Lymphocyte count (μL) | 2.505 | 1.45-4.31 | **0.0009** | 1.944 | 1.068-3.536 | **0.0294** |
| Monocyte count (μL) | 6.006 | 1.40-25.71 | **0.0157** | 2.166 | 0.297-15.765 | 0.4453 |

Abbreviations: IL 4; Interleukin 4; Interleukin 10;HB; Hemoglobin, OR; Odd Ratio; CI, confidence intervals.

**Table 4. Immunological and hematological factors associated with schistosomiasis adult participants.**

| Variable | Crude analysis | | p-value | Adjusted analysis | | p-value |
|---|---|---|---|---|---|---|
| | OR | 95%CI | | AOR | 95%CI | |
| **Sex** | 1.136 | 0.37-3.42 | 0.8197 | 0.610 | 0.079-4.678 | 0.6352 |
| **IL 4 (pg/ml)** | 1.008 | 1.00-1.01 | **< 0.0001** | 1.007 | 1.003-1.012 | **0.0011** |
| **IL 10 (pg/ml)** | 1.00 | 0.99-1.00 | 0.3231 | | | |
| **Eosinophils count (µL)** | 11.10 | 2.60-47.43 | **0.0011** | 12.265 | 0.726-207.125 | 0.0822 |
| **HB (g/dl)** | 0.76 | 0.59-0.96 | **0.0238** | 0.734 | 0.506-1.065 | 0.1043 |
| **Lymphocyte count (µL)** | 1.89 | 1.15-3.12 | **0.0118** | 2.956 | 1.186-7.364 | **0.0199** |
| **Monocyte count (µL)** | 3.52 | 1.14-10.83 | **0.0283** | 0.740 | 0.100-5.468 | 0.7687 |

Abbreviations: IL 4; Interleukin 4; IL-10, Interleukin 10;HB; Hemoglobin, OR; Odd Ratio; CI, confidence intervals.

involvement of lymphocytes across all age groups, suggesting distinct but overlapping immune mechanisms in the host response to schistosomiasis.

The age-dependent significance of IL-4 in our multivariate models presents particularly intriguing findings. While IL-4 showed no independent association in pediatric participants (AOR 1.001, p=0.0715), it emerged as a strong associate of schistosomiasis in adults (AOR 1.007, p=0.0011). This pattern is in agreement with the current understanding of $T_H2$ immune polarization in helminth infections but suggests this response becomes more pronounced with age and likely with chronicity of infection [17]. The stronger $T_H2$ response in adults may reflect immunological memory and repeated antigen exposure, consistent with studies demonstrating enhanced IL-4 production in chronic schistosomiasis patients [18–20]. This finding has important implications for understanding disease progression, as the $T_H2$ response, while initially protective, may contribute to the fibrotic pathology characteristic of advanced schistosomiasis. The absence of this association in children could indicate either a less robust $T_H2$ polarization or the predominance of other immune mechanisms in early infection.

The lack of a significant association between IL-10 levels and schistosomiasis in our multivariate analysis, despite significance in univariate analysis, needs careful consideration. IL-10 is a key regulatory cytokine in helminth infections, known for its role in modulating excessive immune responses and limiting tissue damage. However, its production is highly context-dependent, influenced by factors such as the stage of infection, host immune status, and co-infections. In schistosomiasis, IL-10 is often elevated during chronic stages to counterbalance pro-inflammatory responses, but this increase may not always correlate directly with infection status when controlling for confounders. Our findings suggest that IL-10 may act more as a secondary regulator than a primary indicator of infection. This pattern is consistent with observations in other parasitic diseases, where IL-10 functions to maintain immune homeostasis rather than serve as a reliable marker of active infection.

Lymphocyte counts remained significantly elevated in infected individuals across both age groups after multivariate adjustment. This consistent finding strongly supports the central role of lymphocyte-mediated immunity in schistosomiasis, corroborating numerous studies demonstrating T-cell activation in response to schistosome antigens [21,22]. The higher odds ratio in adults may reflect either greater antigenic stimulation over time or age-related differences in immune regulation. Current literature suggests that while lymphocyte activation is crucial for parasite control, excessive or dysregulated responses may contribute to the granulomatous inflammation characteristic of chronic infection [5,23,24]. Our findings extend this understanding by demonstrating that lymphocyte involvement persists across the age spectrum, though potentially with different functional consequences in pediatric versus adult patients.

Several variables that showed significance in univariate analysis failed to maintain independent associations in our multivariate models. The lack of association with sex after adjustment challenges some previous reports of gender-based

differences in schistosomiasis susceptibility. While biological factors like hormone-mediated immune modulation have been proposed to explain sex differences, our findings contradict these assertions [25]. This has important implications for control programs, emphasizing the need for gender-neutral prevention strategies.

The non-significance of IL-10 in adjusted models presents an interesting paradox, given its well-established role as a regulatory cytokine in helminth infections [26,27]. While IL-10 levels differed significantly between infected and uninfected individuals in univariate analysis, this association disappeared after controlling for other factors. This may suggest that IL-10 production in schistosomiasis is secondary to other immune processes rather than being an independent determinant of infection status. Alternatively, the timing of sample collection relative to infection may have influenced our ability to detect regulatory effects, as IL-10 dynamics can vary considerably during the course of infection.

Similarly, eosinophil counts, while significantly elevated in infected individuals in univariate comparisons, did not maintain independent associations in multivariate models. This finding challenges the traditional view of eosinophils as central effectors in anti-schistosome immunity [28,29]. One possible explanation is that eosinophilia in schistosomiasis may be primarily driven by IL-4 and other $T_H2$ cytokines, making it a secondary rather than primary marker of infection when these other factors are considered, these has been previously proposed by a number of researchers and the finding in this present study supports this theory [6,30,31]. This has important diagnostic implications, suggesting that eosinophil counts alone may have limited value as a screening tool in endemic areas.

The hematological findings, particularly the consistent association between schistosomiasis and reduced hemoglobin levels, reinforce the significant burden of anemia in schistosomiasis-endemic populations. While this association was somewhat attenuated in multivariate models, it's a clinically important finding. The mechanisms linking schistosomiasis to anemia are likely multifactorial and include both direct blood loss and inflammation-mediated suppression of erythropoiesis [32,33]. The public health implications of this finding are substantial, as they highlight the need to integrate schistosomiasis control with anemia management programs, particularly in pediatric populations where the developmental consequences of anemia can be most severe.

Our study findings carry significant implications for both the scientific understanding of schistosomiasis immunopathogenesis and the practical management of the disease in endemic settings like Zambia. From a scientific standpoint, the age-stratified immunological responses, mainly the differential role of IL-4 and consistent lymphocyte involvement emphasize the complexity and dynamism of the host-parasite interaction over time. Our study contributes to the growing body of knowledge suggesting that the immune landscape of schistosomiasis is not static but evolves with age and exposure history, therefore necessitating longitudinal studies to gain insight into these temporal dynamics. Clinically and programmatically, our results suggest that interventions may need to be age-specific, with pediatric cases potentially benefiting from strategies targeting early inflammatory responses, while on the other hand, adult cases may require therapeutic approaches that modulate chronic $T_H2$-driven pathology. Furthermore, the attenuation of traditional markers such as eosinophilia and IL-10 in multivariate analysis highlights the complex interplay between immune effectors and regulatory pathways in schistosomiasis. Our findings suggest that eosinophilia, long regarded as central in helminth infections, may be largely dependent on upstream $T_H2$ cytokines, chiefly IL-4, rather than being an independent indicator of infection status. This reinforces the notion that eosinophil elevation in schistosomiasis may be a downstream consequence of IL-4, driven immune polarization, rather than a primary immune event as previously held. As such, reliance on eosinophil counts for diagnosis or disease monitoring in endemic regions may be inadequate without concurrent assessment of $T_H2$ responses. These insights inform the refinement of diagnostic strategies and have important implications for vaccine development. Targeting IL-4 and other $T_H2$-associated pathways could offer a strategic advantage in modulating host immunity to achieve protective, non-pathogenic responses. Understanding the temporal and functional relationships between cytokines and effector cells like eosinophils may guide the design of vaccines that mimic natural, protective immunity without exacerbating pathology. Ultimately, our study underscores the value of context-specific immunological data in developing more precise, stage-tailored, and effective interventions for the prevention and control of schistosomiasis.

## Strengths and limitations

A notable strength of this research work is the specificity of the area under consideration, the Mulobezi District; an area rarely investigated in relation to schistosomiasis studies. This localized approach enables accurate tracking of the progression of the disease in the community thus offering information for public health measures. In addition, the expansion of endpoints toward biomarkers of immunologic activity, such as interleukin-4 and eosinophil count, which cannot be obtained by conventional histological examination, makes the study valuable from both clinical and immunologic standpoints. With multivariable logistic regression analysis, potential confounding factors are mutually adjusted, which offers relatively accurate results concerning the relationships between threats and schistosomiasis. Finally, the data thus obtained enlighten environmental and behavioral determinants prevailing over schistosomiasis infection and transmission and may hence be useful in restructuring preventive measures towards reduced disease load.

However, several limitations must be considered when interpreting the results. First, the relatively small sample size (143 participants) limits the statistical power of the study and may reduce the generalizability of the findings to other regions. The cross-sectional design of the study prevents the establishment of causality, as it captures a single point in time rather than following participants over time to assess changes in infection status and immune responses. This limitation is particularly important when interpreting immune markers like IL-4 and eosinophils, as their levels can fluctuate during different stages of infection.

Another limitation is the reliance on single measurements of laboratory markers such as IL-4 and eosinophils may not fully capture the dynamic nature of the immune response in schistosomiasis. Repeated measures over time would provide a more accurate picture of immune activity.

Additionally, unmeasured confounders, such as genetic factors, and nutritional status, could influence the associations observed in the study. Variations in environmental factors, such as seasonal changes in water quality or snail populations, were not accounted for, which may affect infection rates and immune responses. These therefore need to be considered in future larger studies to ascertain their influence in the schistosomiasis outcomes.

## Conclusion

Our study demonstrates distinct but overlapping immune profiles associated with schistosomiasis in pediatric and adult populations. The age-dependent significance of IL-4 suggests the maturation of the $T_H2$ response with prolonged exposure, while the lymphocyte involvement in both populations highlights the central role of adaptive immunity in the two age groups. These findings contribute to our understanding of schistosomiasis immunopathogenesis and may inform the development of more targeted approaches to diagnosis and surveillance in endemic areas. Future research should focus on the longitudinal assessment of these immune parameters in relation to infection dynamics and treatment outcomes.

## Methods

### Ethics statement

Before the study, approval and permission were obtained from the National Health Research Authority (NHRA) on 11th January 2024 with Reference number: **NHRA-825/04/01/2024,** Mulungushi University School of Medicine and Health Sciences Ethics Committee (MUHSREC) on 26th December 2023 with Reference number: **SMHS-MU4-2023-031** and Mulobezi District Health Office (DHO) on 8th February 2024. All participants were required to sign an informed consent form. Written informed consent form was obtained from all participants. We ensured that the confidentiality of the data and study participants' privacy was maintained. The study's purpose, procedures, potential risks, and benefits were explained to the participants in a language they understood, and their participation was entirely voluntary.

## Study design and setting

This cross-sectional study was conducted in Mulobezi District, located in Western Zambia. The district is predominantly rural, with most communities relying on river water for daily activities. The study included participants from Machile, Sichili, and Mulobezi communities. The study enrolled a total of 143 participants, 92 females and 51 males. These were stratified into two age groups, pediatric (<15 years) and adults (≥15 years).

## Eligibility criteria

The study included participant aged 5–55 who had resided in Mulobezi District for more than three months. Pregnant women, individuals with severe underlying health conditions such as advanced HIV disease and autoimmune disease.

## Variables

The outcome variable was schistosomiasis infection classified as binary (yes or no) and the independent variables that were considered were age, sex (male or female), immune profile (IL4, IL-10) and blood tests: Hemoglobin, eosinophils count, neutrophils count, lymphocyte count, and monocyte count.

## Data collection

Participants provided urine samples and completed a questionnaire collecting demographic data. Each participant provided 2 blood samples and one urine sample. Blood samples were collected via standard venipuncture into sterile vacutainer tubes, while urine samples were obtained using the midstream clean-catch method. Blood samples were analyzed for Full blood count and serological markers using enzyme-linked immunosorbent assay (ELISA), and urine samples were analyzed for Schistosoma hematobium using microscopy. Urine samples were collected in 50 mL pre-labelled pre-labeled containers and transported within 7 hours in cool boxes to Sichili Mission Hospital for laboratory analysis. Blood samples were collected in labeled containers, with whole blood stored and transported to Metropolis laboratory for a full blood count analysis, while serum from the red-top containers was sent to TDRC Ndola for IL-4 and IL-10 analysis using ELISA. Urine microscopy was performed by centrifuging 10 mL the collected urine samples at 2,000 rpm for 5 minutes. The sediment was then examined under 10x and 40x magnification for Schistosoma haematobium ova. Standard laboratory procedures were followed throughout sample handling and analysis to maintain data integrity.

## Cytokine quantification by ELISA

Plasma levels of IL-4 and IL-10 were measured using human sandwich ELISA kits from Elabscience Biotechnology Inc. (Houston, TX, USA). IL-4 was quantified using kit E-EL-H0101 (sensitivity: 18.75 pg/mL; range: 31.25–2000 pg/mL), while IL-10 was assessed using kit E-EL-H0103 (sensitivity: 0.94 pg/mL; range: 1.56–100 pg/mL). Both assays followed the manufacturer's instructions, used 100 µL of plasma, and were completed in ~3.5 hours. No significant cross-reactivity or interference was reported, and intra-assay variability was <10%. Each ELISA run included commercially available positive and negative controls to ensure assay reliability, reproducibility, and accuracy in line with standard quality control procedures. Absorbance was read at 450 nm, and cytokine concentrations were derived from standard curves.

## Data analysis

The data was analyzed using StatCrunch data analysis software. Data cleaning and coding were conducted using a codebook. Descriptive statistics were used to provide summary statistics for demographic variables and clinical characteristics, schistosomiasis infection status, and other relevant variables. Measures of central tendency (e.g., median) and dispersion (e.g., interquartile range) were calculated for continuous variables, while frequencies and percentages were reported for categorical variables.

To test for Normality Shapiro Wilk's test and histograms were used. Mann-Whitney test was used to compare the median between two groups with not normally distributed data.

The Chi-square Test was used to assess whether there is a significant relationship between two categorical variables. Univariable and multivariable logistic regression was used to examine the association of independent variables with schistosomiasis infection. Statistical tests were performed using two-sided tests to determine the significance of the data, with a significance level set at $p < 0.05$.

## Supporting information

**S1 File.  STROBE - A checklist to aid reporting.**
(DOCX)

**S1 Data.  Data that underlies this paper.**
(XLSX)

## Acknowledgments

The authors would like to thank Mulobezi District health office, and Sichili Mission hospital Management for having granted permission to conduct the study at Sichili Mission hospital laboratory. We are also grateful to the Mulobezi District laboratory coordinator, Adrian Chiyokoma, Chanda Kalebwe and our assistant principal investigators Mwamba Joseph and Kelvin Solochi for their continued support during data collection process. We also thank the Team at Livingstone Center for Prevention and Translational Science (LCPTS) for their support.

## Author contributions

**Conceptualization:** Tresford Lwanga, Sepiso K. Masenga.

**Data curation:** Tresford Lwanga, Lweendo Muchaili, Gift C. Chama, Lukundo Siame, Cornelius Simutanda, Situmbeko Liweleya, Bislom C. Mweene, Sydney Mulamfu, Sepiso K. Masenga.

**Formal analysis:** Tresford Lwanga.

**Methodology:** Tresford Lwanga.

**Project administration:** Tresford Lwanga, Sepiso K. Masenga.

**Resources:** Tresford Lwanga.

**Software:** Sepiso K. Masenga.

**Supervision:** Situmbeko Liweleya, Bislom C. Mweene, Sydney Mulamfu, Sepiso K. Masenga.

**Validation:** Sepiso K. Masenga.

**Visualization:** Sepiso K. Masenga.

**Writing – original draft:** Tresford Lwanga, Lweendo Muchaili, Gift C. Chama, Lukundo Siame, Cornelius Simutanda, Situmbeko Liweleya, Bislom C. Mweene, Sydney Mulamfu, Sepiso K. Masenga.

**Writing – review & editing:** Tresford Lwanga, Lweendo Muchaili, Gift C. Chama, Lukundo Siame, Cornelius Simutanda, Situmbeko Liweleya, Bislom C. Mweene, Sydney Mulamfu, Sepiso K. Masenga.

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
