## [Decision Letter · Decision Letter 0]

Mar 27 2025

Immunological biomarkers and correlates of schistosomiasis in Mulobezi District: a cross-sectional study

Dear Dr. Masenga,

Thank you for submitting your manuscript to PLOS Neglected Tropical Diseases. After careful consideration, we feel that it has merit but does not fully meet PLOS Neglected Tropical Diseases's publication criteria as it currently stands. Therefore, we invite you to submit a revised version of the manuscript that addresses the points raised during the review process.

Please submit your revised manuscript within 60 days Mar 27 2025 11:59PM. If you will need more time than this to complete your revisions, please reply to this message or contact the journal office at plosntds@plos.org. Please include the following items when submitting your revised manuscript:

We look forward to receiving your revised manuscript.

Kind regards,

Feng Xue, Ph.D.

Guest Editor

Jong-Yil Chai

Section Editor

Shaden Kamhawi

co-Editor-in-Chief

Paul Brindley

co-Editor-in-Chief

**Journal Requirements:**

1) Thank you for including an Ethics Statement for your study. For child participants, please include:

i) A statement that formal consent was obtained (must state whether verbal/written) from the parent/guardian OR the reason consent was not obtained (e.g. anonymity). 

2) Please upload the main figure as a separate Figure file in .tif or .eps format. For more information about how to convert and format your figure files please see our guidelines: 

3) We have noticed that you have uploaded Supporting Information files, but you have not included a complete list of legends. Please add a full list of legends for your Supporting Information files after the references list.

Potential Copyright Issues:

i) Figure 1. Please confirm whether you drew the images / clip-art within the figure panels by hand. If you did not draw the images, please provide (a) a link to the source of the images or icons and their license / terms of use; or (b) written permission from the copyright holder to publish the images or icons under our CC BY 4.0 license. Alternatively, you may replace the images with open source alternatives. See these open source resources you may use to replace images / clip-art:

**Comments to the Authors:**

**Please note that one of the reviews is uploaded as an attachment.**

**Reviewers' Comments:**

Reviewer's Responses to Questions

**Key Review Criteria Required for Acceptance?**

**Methods**

-Are the objectives of the study clearly articulated with a clear testable hypothesis stated?

-Is the study design appropriate to address the stated objectives?

-Is the population clearly described and appropriate for the hypothesis being tested?

-Is the sample size sufficient to ensure adequate power to address the hypothesis being tested?

-Were correct statistical analysis used to support conclusions?

-Are there concerns about ethical or regulatory requirements being met?

Reviewer #1: See comments below

Reviewer #2: 1.The inclusion criteria for study participants were not rigorous enough. For example, previous infections are not considered, and previous infections may be important factors that affect immune markers.

2.The details of ELISA were not provided.

3.Why did you only choose IL-4 and IL-10?

4.Did you think the factors you investigated be enough?

5.Relative to the high prevalence of schistosomiasis, the inclusion of study participants is insufficient, and the results are less convincing.

Reviewer #3: (No Response)

Reviewer #4: The study is appropriately designed and methodology supports the objectives and results. The author's have addressed the limitation of the study and sample size which is important for the conclusion of the finding.

Reviewer #5: In the methods section of the abstract (and line 168), please specify whether your tests were one (> or <) or two-sided.

In the methods section of the abstract, please consider replacing “A logistic regression model (bivariate and multivariable) was used to …” with “Unadjusted and adjusted logistic regression models were use to …” or “Simple and multiple logistic regression models were used to …” In lines 167-168, you write, “Univariable and multivariable logistic regression was used to examine the factors …” This is fine, but you need to be consistent in your usage of these terms (correct lines 219-221 in the results section because you alternate between -ate and -able suffixes and they mean different things).

Did you run a sample size calculation based on accurate assumptions? Can you provide a power analysis? It seems like you included a lot of variables in your multivariable model given the small sample size.

Reviewer #6: -Are the objectives of the study clearly articulated with a clear testable hypothesis stated?

Yes

-Is the study design appropriate to address the stated objectives?

Yes, but I expected the authors to use a greater number of immunological markers and a larger sample size categorized by population type.

-Is the population clearly described and appropriate for the hypothesis being tested?

Yes, it is clearly described, but it is not entirely appropriate.

-Is the sample size sufficient to ensure adequate power to address the hypothesis being tested?

No

-Were correct statistical analysis used to support conclusions?

It is not fully supported due to the sample size and population.

-Are there concerns about ethical or regulatory requirements being met?

No

Please, can you provide bibliographical support for the following:

-Why did you choose that sample size, and how would this limit the conclusions of your study?

-Susceptibility to infection according to the sex and age of the patients.

I suggest adding a figure showing the study area.

I suggest indicating how many blood and urine samples were taken per person. Indicate how the samples were taken, and how the analysis of the samples was performed. Also, mention which were the positive and negative controls in the analysis of the samples.

**Results**

-Does the analysis presented match the analysis plan?

-Are the results clearly and completely presented?

-Are the figures (Tables, Images) of sufficient quality for clarity?

Reviewer #1: See comments below

Reviewer #2: Although the analysis presented match the analysis plan, it is not appropriate due to the shortcomings of the design. moreover, no novel findings were produced from this study.

Reviewer #3: (No Response)

Reviewer #4: Yes

Reviewer #5: In the Results section of the abstract, you write, “The prevalence of schistosomiasis in the study population was 39.2% (95% CI).” The values for the 95% CI are missing, so please insert them.

Throughout the results section of the abstract, consider including the p-values or the 95% CIs. Including both is redundant.

In the results section of the abstract (for example), you write, “… river as a source of water for drinking and domestic use was significantly associated with an increased likelihood of schistosomiasis … and the presence of symptoms related to schistosomiasis greatly increased the likelihood of infection …” OR is not a probability, so consider avoiding using the “likelihood” terminology. Instead, just say the odds were higher (or lower). As a reminder, likelihood is related to risk and relative risk, which are probabilities. Throughout the paper, please avoid using “likelihood” unless you plan to use relative risk instead of odds ratios. Instead, just say “higher odds” or “lower odds” here.

In the Synopsis, you write, “We found a significant proportion of 39% of children and adults infected with Schistosoma haematobium.” Consider replacing “significant” with “substantial” so as not to be confused with statistical significance.

Line 120: Replace “march” with “March” and insert the last day for June 2024.

Did you control/adjust for community in your multivariable logistic regression model?

In your multivariable logistic regression model, did you consider any two-way interactions/effect modifications?

What did you include as symptoms related to schistosomiasis? Would there be any benefit to examining them separately?

Throughout the paper, please be consistent in the way you present dates (day, month, year or month, day, year), e.g., lines 120, 171, 173, 174.

In the results section, be consistent in if/how you present the IQR text—as (Q1, Q3) or the difference Q3-Q1—personal preference for the former. Also, be consistent in how you present summary stats for the categorical variables—personal preference for frequency (percent).

In Tables 1 and 2 of the results section, you report only n=1 employed participant. It seems odd to identify this as an occupation variable. n=94 are in school, so there seems to be overlap with the education variable. Consider combining these two variables, given your small sample size, in your models. Consider combining secondary and tertiary education in your models, too. Also, consider combining rarely and never contact with contaminated water in your models. Also, replace “domestic us” with “domestic use” in table 1.

In Table 1, what constitutes “protective measures” here?

What were the other medical conditions that n=15 participants were diagnosed with?

In Table 2, you have some cells with 0-2 participants. Consider collapsing categories for these variables before running statistical tests and fitting models.

In Table 2, the p-value for “treated with schistosomiasis” is incorrect given the cell counts. Please double check all your results. It should be <0.0001.

In your multivariable model (table 3), you use a lot of variables given the small sample size. Please run a power analysis.

Reviewer #6: -Does the analysis presented match the analysis plan?

The analysis presented does not fully match the analysis plan. In the methods, they mention the use of Chi-square, but in the results, there is no value for it. Therefore, I suggest indicating the Chi-square values. Also, I suggest indicating the results of normality tests used. Please.

-Are the results clearly and completely presented?

I suggest representing the relevant results in the tables using plots. Also, I recommend removing Figure 1 if it is not a result of the original research or conceptual interpretation of the discussion process carried out in the paper. In addition, it is necessary to add the reference(s) that allowed this figure to be constructed.

-Are the figures (Tables, Images) of sufficient quality for clarity?

Yes, but keep in mind representing the relevant results on plots. Please.

**Conclusions**

-Are the conclusions supported by the data presented?

-Are the limitations of analysis clearly described?

-Do the authors discuss how these data can be helpful to advance our understanding of the topic under study?

-Is public health relevance addressed?

Reviewer #1: See comments below

Reviewer #2: although you discussed the limitations of your study, it needs significant improvements from design and analysis.

Reviewer #3: (No Response)

Reviewer #4: Yes

Reviewer #5: Yes

Yes

Yes

Yes

Reviewer #6: -Are the conclusions supported by the data presented?

Please. Limit the conclusion according to the scope and limitations of your research.

-Are the limitations of analysis clearly described?

Yes, in the discussion, but no in the conclusions.

-Do the authors discuss how these data can be helpful to advance our understanding of the topic under study?

Yes

-Is public health relevance addressed?

Yes

Please. Add the year of publications named with the author name et al., eg. On discussion the lines following:

244 “Tembo et al.”

251 “Olorunlana et al.”

**Editorial and Data Presentation Modifications?**

Reviewer #1: See comments below

Reviewer #2: (No Response)

Reviewer #3: (No Response)

Reviewer #4: Minor Revision

Reviewer #5: (No Response)

Reviewer #6: (No Response)

**Summary and General Comments**

Reviewer #1: This manuscript reported the association between potential risk factors and schistosomiasis in Mulobezi District based on a cross-sectional study. The topic falls into the scope of the journal. The research is well presented. However, I have several questions before it can be published.

1. Title: ‘correlates’ looks weird. Could be ‘risk factors’?

2. Abstract: Line 37- 38. “A logistic regression model (bivariate and multivariable) was used to…”, which is inconsistent with the method (Univariable and

multivariable logistic regression) described in the method section and result section

3. Introduction: Please briefly provide some background on biomarkers for Schistosomiasis

4. Methods: can the authors obtain some other environmental and biological data, such as land cover, meteorological data, snail population, water exposure data?

5. In the regression model, how did the authors select the final model? Is the multicollinearity issue considered?

6. Results: what are the associations for different age groups (children, elderly). It is likely that young children and elderly population are vulnerable to this infection.

7. What are confounders considered in the multivariable logistic regression?

8. Discussion: The authors found the significant association between using river water likelihood of schistosomiasis infection. Can the authors elaborate what the pathways to expose the parasite through river water, such as drinking, recreational activities, or clothe and hand washing? Is the water boiled or treated before drinking?

9. Can the authors discuss the potential influence on the results by confounders?

10. Some associations are staggering, for example: “the presence of symptoms related to schistosomiasis greatly increased the likelihood of infection (AOR = 108.07, 95% CI: 11.24-1037.61, p < 0.001)”. Can the author explain what the possible reasons are?

Reviewer #2: (No Response)

Reviewer #3: (No Response)

Reviewer #4: Minor Comments

-Line 96; the sentence is incomplete and uncomprehensive.

-Line 154; Please provide the code book reference and some details.

-Line 194; The eosinophil count representation is confusing, either correct or please provide detail.

In the discussion, line 242-243, the author's state about the prevalence of disease is more in females in comparison to males, and later provide the describe the reasoning about it. The sample size is too small for the statement and probably the statement could be described with the reasoning together at the same place.

The study is well planned and results are supported by appropriate statistical analysis. This kind of studies are important for the epidemiological mapping, containment of the disease and its timely treatment by providing more information to the affected people.

Reviewer #5: Overall, the manuscript is well-written and the statistical methods are correct. Please double check spelling and punctuation throughout the manuscript. Make sure your results are correct. There are some typos that may change your results/interpretations.

Reviewer #6: The paper titled “Immunological biomarkers and correlates of schistosomiasis in Mulobezi District: a cross-sectional study” interestingly presents a study that combines the use of clinics and epidemiological traits to find relationships between immune response and schistomatosomiasis in Mulobezi, a location with no information on the subject. I applaud the tenacity in obtaining these samples. However, the study lacks an adequate sample size, a categorization of the study population, and an exhaustive environmental sampling. In addition, more in-depth cellular and molecular studies are needed to propose an immune response mechanism.

PLOS authors have the option to publish the peer review history of their article (what does this mean? ). If published, this will include your full peer review and any attached files.

**Do you want your identity to be public for this peer review?** For information about this choice, including consent withdrawal, please see our Privacy Policy .

Reviewer #1: No

Reviewer #2: No

Reviewer #3: No

Reviewer #4: No

Reviewer #5: No

Reviewer #6: No

**Figure resubmission:**

**Reproducibility:**



---

## [Decision Letter · Decision Letter 1]

Dear Prof. Masenga,

We are pleased to inform you that your manuscript 'Expression of Interleukin-4 in Schistosomiasis is influenced by Age' has been provisionally accepted for publication in PLOS Neglected Tropical Diseases.

Best regards,

Feng Xue, Ph.D.

Guest Editor

Jong-Yil Chai

Section Editor

Shaden Kamhawi

co-Editor-in-Chief

Paul Brindley

co-Editor-in-Chief

Reviewer's Responses to Questions

**Key Review Criteria Required for Acceptance?**

**Methods**

-Are the objectives of the study clearly articulated with a clear testable hypothesis stated?

-Is the study design appropriate to address the stated objectives?

-Is the population clearly described and appropriate for the hypothesis being tested?

-Is the sample size sufficient to ensure adequate power to address the hypothesis being tested?

-Were correct statistical analysis used to support conclusions?

-Are there concerns about ethical or regulatory requirements being met?

Reviewer #1: (No Response)

Reviewer #2: The major drawbacks for this paper is the low sample size of 143 and the simple age grouping of <15 and >15, which may limit the statistical power and generalizability of the findings reported.

Reviewer #3: (No Response)

Reviewer #4: The study design address the undertaken hypothesis and the results are supported by proper data and statistical analysis.

Reviewer #5: Ok

**Results**

-Does the analysis presented match the analysis plan?

-Are the results clearly and completely presented?

-Are the figures (Tables, Images) of sufficient quality for clarity?

Reviewer #1: (No Response)

Reviewer #2: The design will limit the statistical power and generalizability of the findings reported.

Reviewer #3: (No Response)

Reviewer #4: Yes

Reviewer #5: Ok

**Conclusions**

-Are the conclusions supported by the data presented?

-Are the limitations of analysis clearly described?

-Do the authors discuss how these data can be helpful to advance our understanding of the topic under study?

-Is public health relevance addressed?

Reviewer #1: (No Response)

Reviewer #2: The findings from this study is only limited to schistosomiasis caused by Schistosoma haematobium or any other

Reviewer #3: (No Response)

Reviewer #4: Yes

Reviewer #5: Ok

**Editorial and Data Presentation Modifications?**

Reviewer #1: (No Response)

Reviewer #2: (No Response)

Reviewer #3: (No Response)

Reviewer #4: Accept

Reviewer #5: The authors have addressed all my questions and/or concerns. I have no further comments or edits.

**Summary and General Comments**

Reviewer #1: The authors have addressed my concerns. No additional comments

Reviewer #2: (No Response)

Reviewer #3: (No Response)

Reviewer #4: (No Response)

Reviewer #5: The authors have addressed all my questions and/or concerns. I have no further comments or edits.

PLOS authors have the option to publish the peer review history of their article (what does this mean? ). If published, this will include your full peer review and any attached files.

**Do you want your identity to be public for this peer review?** For information about this choice, including consent withdrawal, please see our Privacy Policy .

Reviewer #1: **Yes: ** Jianyong Wu

Reviewer #2: No

Reviewer #3: No

Reviewer #4: No

Reviewer #5: No

---

## [Editor Report · Acceptance letter]

Dear Prof. Masenga,

We are delighted to inform you that your manuscript, "Expression of Interleukin-4 in Schistosomiasis is influenced by Age," has been formally accepted for publication in PLOS Neglected Tropical Diseases.

Best regards,

Shaden Kamhawi

co-Editor-in-Chief

Paul Brindley

co-Editor-in-Chief
